# Maximizing Recovery of Paenibacillin, a Bacterially Produced Lantibiotic, Using Continuous Foam Separation from Bioreactors

**DOI:** 10.3390/foods11152290

**Published:** 2022-07-31

**Authors:** Emily P. Campbell, David R. Kasler, Ahmed E. Yousef

**Affiliations:** 1Department of Food Science and Technology, The Ohio State University, Columbus, OH 43210, USA; campbell.2179@buckeyemail.osu.edu (E.P.C.); kasler.11@osu.edu (D.R.K.); 2Department of Microbiology, The Ohio State University, Columbus, OH 43210, USA

**Keywords:** antimicrobial peptides, paenibacillin, *Paenibacillus polymyxa*, bioreactor, foam separation, response surface methodology

## Abstract

Industrial production of paenibacillin, and similar rare antimicrobial peptides, is hampered by low productivity of the producing microorganisms and lack of efficient methods to recover these peptides from fermentor or bioreactor end products. Preliminary data showed that paenibacillin was preferentially partitioned in foam accumulated during growth of the producer, *Paenibacillus polymyxa*, in aerated liquid media. This research was initiated to improve the production and recovery of paenibacillin in bioreactors by maximizing partitioning of this antimicrobial agent in the collected foam. This was completed through harvesting foam continuously during paenibacillin production, using modified bioreactor, and optimizing bioreaction conditions through response surface methodology (RSM). During initial screening, the following factors were tested using 400 mL inoculated media in 2 L bioreactors: medium (tryptic soy broth, TSB, with or without added yeast extract), airflow (0 or 0.8 L/min; LPM), stir speed (300 or 500 revolution/min; RPM), incubation temperature (30 or 36 °C), and incubation time (16 or 24 h). Results showed that airflow, time, and stir speed had significant effects (*p* < 0.05) on paenibacillin recovery in the collected collapsed foam (foamate). These factors were varied together to follow the path of steepest assent to maximize paenibacillin concentration. Once the local maximum was found, RSM was completed with a central composite design to fine-tune the bioreaction parameters. The optimization experiments proved that the significant parameters and their optimal conditions for paenibacillin concentration in the foam were: incubation at 30 °C for 23 h with airflow of 0.95 LPM, and agitation speed of 450 RPM. These conditions increased paenibacillin concentration, predicted by RSM, from 16 µg/mL in bioreaction without foam collection to 743 µg/mL collected in foamate. The optimized conditions also almost doubled the yield of paenibacillin measured in the foam collected from a bioreaction run (12,674 µg/400 mL bioreaction) when compared to that obtained from a run without foam collection (6400 µg/400 mL bioreaction). Results of this study could improve the feasibility of commercial production and downstream processing of paenibacillin and similar novel antimicrobial peptides. Availability of such peptides will eventually help in protecting perishable products against pathogenic and spoilage bacteria.

## 1. Introduction

Paenibacillin is a promising lanthionine-containing polycyclic antimicrobial peptide (i.e., lantibiotic) produced by *Paenibacillus polymyxa* [1,2,3,4,5]. The lantibiotic is highly water soluble and has relative stability against high temperature and pH change. It is potent against Gram-positive bacteria including species of *Listeria*, *Clostridium*, *Lactobacillus*, *Staphylococcus* and *Bacillus* [4]. The broad antimicrobial coverage and desirable physico-chemical characteristics make paenibacillin a promising antimicrobial candidate for the preservation of food, cosmetics, and other perishable products. The wild producer strain, *P. polymyxa* OSY-DF [4], has been improved and an alternate strain, *P. polymyxa* OSY-EC, that produces paenibacillin on a consistent basis was developed recently [6]. The new strain resulted from screening for wild strain’s spontaneous mutants that are resistant to antibiotics known to target protein synthesis mechanisms. Despite the enhancement achieved through strain selection [6], there is a need for improving the recovery of paenibacillin produced in industrially relevant bioreactions.

Production of metabolites by aerobic microorganisms often involves vigorous stirring and air sparging; this results in the traditionally undesirable foam accumulation in the bioreactor. The typical response of bioreactor operator is to suppress foam formation; however, foaming can be beneficial. Foaming may be utilized to separate and concentrate beneficial proteins produced during aerobic bioreactions. Air introduction during stirring increases air–water interface. Protein molecules with hydrophobic and hydrophilic regions (i.e., with amphiphilic structure) may accumulate in aerated bioreactors at this interface and stabilize the foam [7]. If the bioreaction amphiphilic products include antimicrobial peptides, these also accumulate in the air–water interface, i.e., the foam [8]. Migration of these antimicrobial agents to bioreactor foam may indirectly decrease product feedback inhibition [9,10]. 

Foam separation of amphiphilic antimicrobial molecules has been used successfully to recover nisin produced by *Lactococcus lactis* [9,11,12,13], and surfactin [14] and iturin [10] produced by *Bacillus subtilis*. Most of these studies dealt with foam fractionation after bioreactions were completed, and only few researchers have studied foam collection continuously during bioreactions [8]. Foaming not only concentrates the antimicrobial agents during the bioreaction, but it also increases product yield. Nisin production was increased 36.2% when continuous foam recovery was used, compared to fermentation with no foam collection [8]. 

Research is needed to determine the optimum bioreactor conditions that are necessary for maximizing the fractionation of antimicrobial agents into foamate. Previous studies included optimizing the separation of bioactive peptides from hydrolyzed proteins by response surface methodology (RSM) [15] and optimized separation of nisin by foam fractionation through one-factor-at-a-time methodology [9,11,12]. However, foam-separation of antimicrobial agents produced by bacteria during fermentations or bioreactions has not been reported to be optimized by the RSM approach. Such optimization should include determining optimal levels of crucial factors such as pH, temperature, and aeration, through statistical design of experiments (DOE) [16]. DOE-based methods are more complicated than the one-factor-at-a-time approach but are more time and resource efficient. Unless the fermentation or bioreaction was already at maximum productivity, the process can be improved through DOE.

This study was initiated to determine if paenibacillin, a scarcely produced antimicrobial peptide, could be separated efficiently in foam, and if so, could paenibacillin titer and yield be improved in the foamate through manipulation of bioreaction parameters. Therefore, determining conditions that maximize paenibacillin production and recovery in foamate was targeted in the study. When successfully recovered in foamate, downstream processing of paenibacillin would require fewer purification steps and large-scale production would be more economically viable than that encountered in traditional bioreactors. 

## 2. Methods

### 2.1. Microbial Strains and Growth Media 

The newly developed efficient paenibacillin-producing strain, *Paenibacillus polymyxa* OSY-EC [6], was used in this study. *Listeria innocua* ATCC 33090 was used as a sensitive bioassay indicator to determine paenibacillin concentration; this strain has been used previously for measuring anti-Gram-positive activity of paenibacillin-producing culture [1]. Stock cultures of these strains were prepared in 40% glycerol medium and stored at −80 °C until used in these experiments. *P. polymyxa* OSY-EC were cultured in tryptic soy broth (TSB; Becton Dickinson and Company, Franklin Lakes, New Jersey, NJ, USA) or TSB containing 0.6% yeast extract (TSB-YE; Becton Dickinson and Company). The bacterium was transferred in these media twice, with incubation in a shaker incubator (120 rpm) for 24 h at 30 °C. The resulting bacterial culture (~10^8^ CFU/mL) was used to inoculate the respective medium in the bioreactor at 0.1% *v*/*v* level. *L. innocua* ATCC 33090 was cultured in TSB at 37 °C prior to use in paenibacillin bioassay.

### 2.2. Bioreactor Setup

Production of paenibacillin was carried out in a 2 L bioreactor vessel, mounted on a reactor base (VirTis Omniculture; The Virtis Company, Inc., Gardiner, NY, USA) as shown in Figure 1. Atmospheric air was sterilized by a 0.45 µm filter before entering the bioreactor. A water bath was used to pump heated water through metal tubing within the reaction vessel to obtain the desired incubation temperature. All reactions were completed using 400 mL of TSB or TSB-YE, which was dispensed into the reactor vessel. The vessel was sealed, autoclaved, and cooled to ambient temperature before inoculation. Two similar bioreactors were run side by side to increase run capacity. The bioreactions were completed with or without foam collection and results were compared. In the former case, the foam was collected, throughout the incubation, in 1 L flasks connected by tubing to the top of the bioreactor (Figure 1), as described in the next section. After incubation, the spent medium in the vessel or the collected foamate was centrifuged at 10,000× *g* for 7 min at 4 °C to remove cells. The cell-free spent medium or foamate was stored briefly at 4 °C until tested for paenibacillin titer determination.

### 2.3. Foam Collection Design

A conventional fermentor vessel was modified to allow for foam collection (Figure 1). The vessel lid was sealed well with a gasket and clamp. Ports that were typically designed for easy plug removal, such as the inoculation port, were replaced with screw capped openings that hold pressure. The vessel vent was changed to a foam port; it was connected to the top of the foam collection flask. During the bioreaction run, a large volume of foam filled the head space of the vessel and flowed out of the foam port, up the tube, to the top of the foam collection flask. The foam entered the collection flask and collapsed into foamate over time. Excess air left through the vent of the foam collection flask. The foam collection flask also had a bottom port that could be used to return foamate to the main reactor or clamped off when foamate collection was desired. Experiments were completed with or without foam collection. 

### 2.4. Paenibacillin Determination in Bulk Media or Foamates

During the initial screening for optimum bioreaction conditions, amounts of paenibacillin produced in bulk media or foamates were estimated as described previously [1] with modifications. Briefly, aliquots (10 µL) of cell-free bioreaction product were spotted, in duplicate, on a soft TSA (containing 0.75% agar) overlay seeded with *L. innocua* ATCC 33090 and diameters of the areas of inhibition (clearing) were measured after incubating the plates for 24 h at 37 °C. Larger area of inhibition indicated greater concentration of paenibacillin. Before calculating paenibacillin concentration, a dose–response standard curve (Figure 2) was constructed as follows. Stock of purified paenibacillin, prepared as described previously [4], was used in these experiments. Known concentrations of paenibacillin were spotted on indicator-seeded soft agar overlay and plates were incubated, as described previously. Diameters of areas of inhibition were measured and plotted against the corresponding paenibacillin concentrations. The equation that described the line of best fit (Equation (1)) was then used to calculate paenibacillin concentration in bulk media or foamates from the diameter of the corresponding inhibition areas (Figure 2).
(1)Paenibacillin, log(µg/mL)=3.589+4.595×Diameter of inhibition area (mm)

During the final optimization of bioreaction conditions, paenibacillin production was quantified using a microdilution method as described previously [1,17,18], with some modifications. Briefly, the cell-free bulk medium or foamate underwent two-fold serial dilutions and 10 µL aliquots were dispensed onto soft TSA seeded with *L. innocua.* Plates were incubated at 37 °C for 24 h. According to a previous work in this laboratory, the minimum inhibitory concentration of paenibacillin against *L. innocua* is 4 μg/mL [1]. Therefore, the concentrations of paenibacillin were calculated from the most diluted preparation that produced detectable inhibition area on the indicator lawn, as show in Equation (2).
(2)$$Paenibacillin\left(µg/\mathrm{mL}\right)=\frac{4\;µg}{\mathrm{mL}}*\frac{1}{Dilution\;factor\;for\;last\;dilution\;with\;inhibition}$$

### 2.5. Statistical Design to Screen Parameters That Influence Paenibacillin Concentration

A partial factorial screening design (JMP, Version 12.2.0; SAS Institute Inc., Cary, NC, USA) was developed to determine how paenibacillin production was influenced by media composition (TSB or TSB-YE), airflow (0 or 0.8 L/min, LPM), stir speed (300 or 500 revolution/min, RPM), incubation temperature (30 or 36 °C), and incubation time (16 or 24 h), as shown in Table 1. Main effects of different parameters on paenibacillin concentration in bulk medium or foamate were determined as shown in Equation (3),
(3)$$y=\beta_{0}+\beta_{1}x_{1}+\beta_{2}x_{2}+\beta_{3}x_{3}+\beta_{4}x_{4}+\beta_{5}x_{5}+\varepsilon$$
where *y* was paenibacillin concentration, and the independent variables were as follows; *x*_1_ time, *x*_2_ airflow, *x*_3_ stir speed, *x*_4_ microbiological medium, and *x*_5_ incubation temperature. Additionally, *ε* was the model’s error term.

Considering that two bioreactors were run side by side, the bioreactor used also was included in the model (as a variable) if it had a significant effect; this variable is not shown in Equation (3). Factors with a significant effect on the model (*p* value < 0.05) were then varied together to determine the path of steepest assent. Any parameter without a significant effect (*p* value > 0.05) was removed from the model; for further experiments, such parameter was set to its the lowest value to reduce analysis cost.

### 2.6. Response Surface Methodology (RSM) to Improve Paenibacillin Titer

The screening model (Equation (3)) determined the bioreaction parameters with significant effect on paenibacillin concentration; these parameters were studied further to improve the concentration in the bulk medium or foamate. The significant parameters were varied together from the determined optimal step size, and optimal conditions were then fine-tuned with RSM (JMP version 12.2.0; SAS Institute Inc., Cary, NC, USA). A two-level central composite design was used to investigate the interaction of the significant parameters (as determined in the initial screening) near their optimum conditions as shown in Table 2 and Table 3. The parameters tested, and their ranges, were airflow (0–0.8 LPM) and incubation temperature (30–36 °C) in case of bioreactions with no foam collection (Table 2), and airflow (0.8–1.2 LPM), stir speed (450–540 RPM), and incubation time (23–26 h), in case of bioreactions with foam collection (Table 3). Two similar bioreactors were used side-by-side to complete the study; hence, the bioreactors utilized were monitored and included in the model, as a variable, if it had a significant effect. Factors with significant effect only (*p* < 0.05) on paenibacillin concentration in the bulk medium or foamate were included in the response surface model and optimal conditions of these parameters for bioreactions were calculated. 

### 2.7. Bioreactor Capacity Coefficient for Scale-Up

The capacity coefficient was calculated as described in previous publications [19,20]. The airflow and stirring parameters along with volume and dimensions of bioreactor vessel were incorporated into determining the capacity coefficient, which describes the degree of saturation as a function of time. 

## 3. Results and Discussion

In some aerated bioreactions completed in this laboratory, antimicrobial peptides were partitioned more favorably in the condensed foam (foamate) than in the bulk bioreactor medium. To complete such bioreactions successfully, the foamate had to be returned to the bulk medium in the bioreaction vessel. However, considering the ease of collecting the foam and purifying the antimicrobial agent from foamates, this study was initiated to determine conditions that maximize paenibacillin production by *P. polymyxa* OSY-EC and recovery of the antimicrobial compound in foamate. For comparative purposes, experiments on production of paenibacillin in bioreactors with or without foam collection were completed.

### 3.1. Initial Screening of Parameters That Influence Paenibacillin Concentration

The initial screening for paenibacillin concentration in bulk media of bioreactions, completed with no foam collection, covered these parameters: medium composition, incubation temperature and time, air flow, and stir speed (Table 1). Among these parameters, airflow and incubation temperature had significant effect (*p* < 0.05). The relationship between paenibacillin concentration and these significant parameters is described by the following equation (Equation (4)):(4)$$ỳ=7.583-5.083x_{a}-4.571x_{b}$$
where ỳ was paenibacillin titer (µg/mL), $x_{a}$ was airflow (LPM), and $x_{b}$ was incubation temperature (°C). The R^2^ of the model was 0.53 and the standard error of the estimate was 7.28. Therefore, the model did not explain all variability in paenibacillin production, but it allowed determination of the step size required to further improve the titer of the antimicrobial agent. 

The paenibacillin producer, *P. polymyxa*, is an aerobic bacterium that requires oxygen for its optimal growth [21]. Production of paenibacillin by this bacterium also requires aeration of the bioreaction medium [1,4]. This aeration was accomplished in current study through controlling airflow and agitation speed. Therefore, it is plausible that airflow was found to influence paenibacillin production (Equation (4)). *Paenibacillus* spp. grow optimally in the range between 28 °C and 40 °C [21] and paenibacillin production was reported previously at 30 °C and 37 °C [1,4]. Based on the results of the current study, paenibacillin production was favored at 30 °C (Table 1).

When bioreactions were run with foam collection, airflow, stir speed, and incubation time were the significant parameters for paenibacillin accumulation in foamates, as described in following equation:(5)$$ŷ=0.390+0.299x_{1}+0.309x_{2}+0.232x_{3}$$
where ŷ was paenibacillin titer (µg/mL), $x_{1}$ was airflow (LPM), $x_{2}$ was incubation time (h), and $x_{3}$ was stir speed (RPM). The R^2^ value of the model was 0.55; thus, the model did not explain all variability in paenibacillin production. However, the model allowed determination of the step size required to further improve the recovery of the antimicrobial agent.

The significant parameters in this bioreaction affect different aspects of paenibacillin production and foam formation. A sufficient bioreaction time is needed for production threshold of paenibacillin, accumulation of the antimicrobial agent in sufficient amounts, and its migration to the air–water interface. Excessively long incubation time may cause the degradation of accumulated paenibacillin in bioreaction medium [6]. Stir speed and airflow influence foam formation and stability, and therefore affect the ability of foam to form and reach the collection flask [22]. Both methods of air incorporation in bioreaction media (i.e., stirring and air sparing) need to be balanced to provide a suitable amount of dissolved oxygen for optimum bacterial metabolism leading to production of antimicrobial agents. 

Paenibacillin is a secondary metabolite and thus production by *P. polymyxa* OSY-EC commences at the end of the exponential growth phase [6]. Paenibacillin concentration in the foamate was not significantly influenced by the tested variations in growth media and incubation temperatures. Both media tested supported the production of paenibacillin and foam formation; therefore, it is obvious that nutrient concentration was not a limiting factor in production of paenibacillin. The temperature range tested supported growth of the producer and production of paenibacillin in previous studies [1,4]. The parameters with significant effect on paenibacillin concentration were further optimized, as described in subsequent sections, to maximize paenibacillin titer and yield in foamate.

### 3.2. Calculated Step Size to Improve Paenibacillin Concentration

Modifications to bioreaction conditions were calculated to improve paenibacillin production and recovery. In case of bioreactions without foam collection, incremental modifications (i.e., step sizes), calculated from the screening model (Equation (4)), were determined to be a decrease of 1 LMP airflow and a decrease of the incubation temperature by 3 °C. The incremental changes were applied to the optimal conditions found in the initial screening. Several methods for reduction in airflow were attempted to simulate negative airflow: bubbling nitrogen instead of air, bioreaction at 0 LPM air, and addition of chemical foam disrupter (antifoam A; Sigma-Aldrich, St. Louis, MO, USA) at 3 ppm. Applying the calculated steps did not improve the antimicrobial agent production. Therefore, the original bioreaction parameters were considered within the optimal range and were used for RSM. 

For bioreactions with foam collection, step sizes were as follows: increase in airflow by 0.1 LPM, stir speed by 19 RPM, and incubation time by 1 h. The incremental changes were applied to the optimal conditions found in the initial screening. The resulting parameters used for the first step were 1.1 LPM, 519 RPM, and 25 h, respectively. The corresponding paenibacillin titer in foamate was 128 µg/mL; therefore, another incremental improvement step, of the same size, was taken. The resulting parameters for bioreaction were 1.2 LPM, 540 RPM and 26 h, respectively. Paenibacillin titer in foamate did not improve; therefore, the second step parameters were used as the upper limit when RSM was applied to finetune the parameters. 

### 3.3. Response Surface Methodology to Improve Paenibacillin Concentration

A two-level central composite design was used to identify the optimum airflow and incubation temperature for maximum paenibacillin titer in bioreactors with no foam collection (Table 2). The original parameters tested contained the optimal bioreaction conditions. Several replicates were required, at key conditions, due to variability in antimicrobial agent production. The RSM model was defined as follows (Equation (6)):(6)$$ỳ=-579-90.0x_{a}+\left(37.4+2.73x_{a}\right)x_{b}-0.589x_{b}{}^{2}$$
where ỳ was paenibacillin titer (µg/mL), $x_{a}$ was airflow (LPM), and $x_{b}$ was temperature (°C). The R^2^ of the model was 0.48 and the standard error of the estimate was 4.4. The RSM predicted a saddle point with maximum paenibacillin production of 16 µg/mL at incubation temperature of 33 °C and airflow of 0.6 LPM (Figure 3). It can be concluded that a small increase in temperature at a low level of airflow produced the highest paenibacillin titer in bioreactions with no foam collection. The original parameters were fine-tuned; this could facilitate paenibacillin production scaling up from flasks to bioreactors. Other researchers have found similar outcome when scale-up runs were optimized and were able to achieve similar yields to the smaller scale bioreactions [23,24].

Similarly, in bioreactions with foam collection, a two-level central composite design was used to investigate the interaction of the significant parameters near the optimum bioreaction conditions. The parameters, and levels tested, were airflow (0.8, 1 or 1.2 LPM), stir speed (450, 500, 520 or 540 RPM), and incubation time (23, 24, 25 or 26 h) as shown in Table 3. The optimum airflow, stir speed, and incubation time were identified through RSM, as depicted by the following model (Equation (7)): (7)$$ŷ=43801+6675x_{1}-3652x_{2}-3x_{3}-3500x_{1}{}^{2}+74x_{2}{}^{2}+s_{1,2}$$
where ŷ was paenibacillin concentration in foamate (µg/mL), $x_{1\;}$ was airflow (LPM), $x_{2\;}$ was time (h), $x_{3\;}$ was stir speed (RPM), and *s*_1,2_ was the bioreactor used (*s*_1_ = 117; *s*_2_ = −117). The R^2^ value of the model was 0.74. The optimal conditions for concentrating paenibacillin in foam, calculated from Equation (7), were airflow of 0.95 LPM, stir speed of 450 RPM, and bioreaction time of 23 h. Under these conditions, predicted maximum paenibacillin titer was 743 µg/mL (Figure 4). These parameters produced foam rich in paenibacillin and were effective in carrying that foam to the collection flask without diluting with excess spent microbiological medium. It was likely that excessive foam formation dilutes the paenibacillin in the foamate. Conversely, a small dense foam did not produce a large enough volume to be collected and examined. A summary of the conditions tested and optimal bioreaction parameters found is presented in Table 4.

### 3.4. Concentration and Yield Improvements

In bioreactions with no foam collection, the concentration of paenibacillin in bulk media ranged from <4 to 32 µg/mL (Table 2), whereas the RSM model predicted a maximum of 16 µg/mL at optimum bioreaction conditions (Figure 3). During experiments using the central composite design conditions, concentrations above 16 µg/mL were not reached. In contrast, bioreactions with foam collection resulted in paenibacillin concentrations ranging from <4 to 768 µg/mL foamates (Table 3) and the RSM model predicted a maximum of 743 µg/mL at optimum bioreaction parameters (Figure 4). Based on these findings, the RSM models predicted 46.4-fold increase in the concentration of paenibacillin in foamates, compared to that in bulk media of bioreactions completed with no foam collection. 

During the runs with foam collection, no antimicrobial activity was detected in the bulk media (data not shown); this suggests that most, in not all, of the antimicrobial agent in the medium bulk has migrated to the foam. The bioreactor used in runs with foam collection had a significant effect on the antimicrobial titer in foamates (Equation (7)). The two bioreactors were set up as similarly as possible but slight differences such as propeller height and collection tube length could have altered the concentration of the paenibacillin collected. 

The optimization process not only increased the migration of paenibacillin to the foam, but it also increased the yield of the antimicrobial agent. In the 400 mL bioreaction, paenibacillin production reached 12,700 µg as collected in foamate where paenibacillin production without foam removal reached 6400 µg during fine tuning (Table 2 and Table 3). The almost double yield reported here could be due to lowered feedback inhibition [9], allowing the production of paenibacillin to continue uninhibited during the microbial bioreaction. Antimicrobial peptides produced at an earlier stage of microbial growth may be destroyed by degradative enzymes, such as peptide hydrolase, which are released at a later stage of the growth of the same microorganism. For example, biosynthesis of poly(epsilon-L-lysine) by *Streptomyces lydicus* USE-11 was degraded by the producer’s peptide hydrolase [25]. Partitioning of antimicrobial peptides into foam, and separation of the peptide-rich foam from bulk media, not only limits potential feedback inhibition of the producer strain but may also alleviates possible product degradation by the producer’s degradative enzymes.

### 3.5. Application to Other Bioreactors 

Calculating the capacity coefficient allows application of the optimum conditions concluded in this study to other bioreactors. In case of bioreactions with foam collection, the optimal airflow and stir speed parameters were incorporated into the calculation of the capacity coefficient. The capacity coefficient can be compared between different bioreactors of different sizes [20]. In the current study, the capacity coefficient for the optimized conditions was 187 s^−1^. If the current bioreaction is scaled up and the capacity coefficient is held constant, the paenibacillin titer in the formed foam is expected to remain at the high concentration reached in the current study.

Most bioreactors can be modified to collect foam with little difficulty. In this case, low-pressure feed lines may require check valves to prevent foam back-flow. The bioreactor can be easily set up as a continuous-feed or fed-batch reactor. However, selecting the appropriate media and reaction conditions to maintain the microorganism in the correct state of antimicrobial production requires additional investigations. 

## 4. Conclusions 

Paenibacillin was collected and concentrated successfully in bioreaction foamate. The overall paenibacillin yield increased 2 fold, approximately, and the antimicrobial agent titer increased 46.4 fold. The bioreaction parameters required for maximum concentration of paenibacillin in foam were successfully identified. Foam separation of paenibacillin was a solvent-free approach to increase the antimicrobial agent’s titer and production. Successful scale-up to larger bioreactors is predicted under the optimized capacity coefficient of 187 s^−1^. Foam separation of antimicrobial peptides improves the feasibility of application of these peptides in perishable products to protect against spoilage and pathogenic organisms. Increased concentration of paenibacillin in foamate diminishes the need for the expensive concentration and purification steps needed during downstream processing of bioreactor products. Follow up studies on the purity of paenibacillin in the foamate will help determine any additional steps required before application in products. Optimizing foam separation by RSM approach could be implemented in other antimicrobial peptide production studies to increase yield and titer with a foam concentration step. Future work is needed to automate the collection of foamate and prevent its overflow, which can happen if bioreaction conditions are changed inadvertently. Alternatively, a mechanism to condense the foamate as it reaches the collection vessel would be beneficial.

## Figures and Tables

**Figure 1 foods-11-02290-f001:**
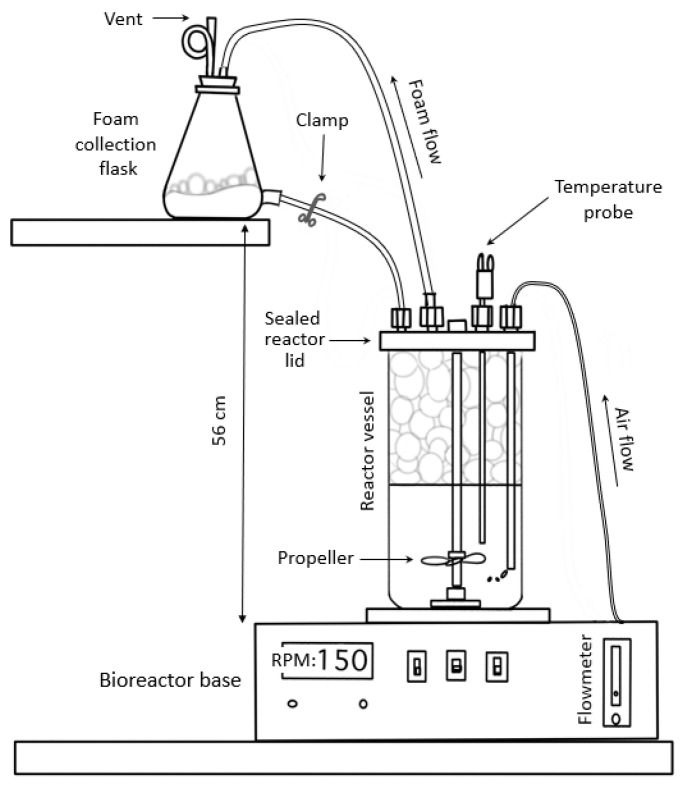
Modified bioreactor setup, customized for separation of foam-containing antimicrobial peptides.

**Figure 2 foods-11-02290-f002:**
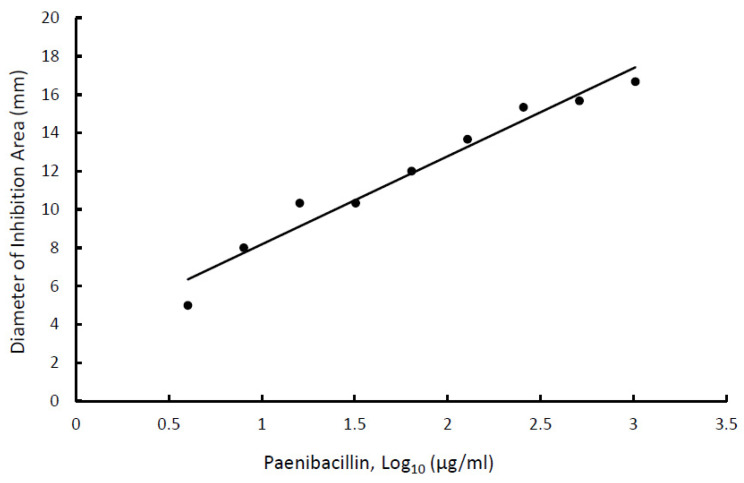
Standard curve for paenibacillin concentration [log_10_ (µg/mL)] vs. diameter (mm) of the area of inhibition of the indicator bacterium, *Listeria innocua*.

**Figure 3 foods-11-02290-f003:**
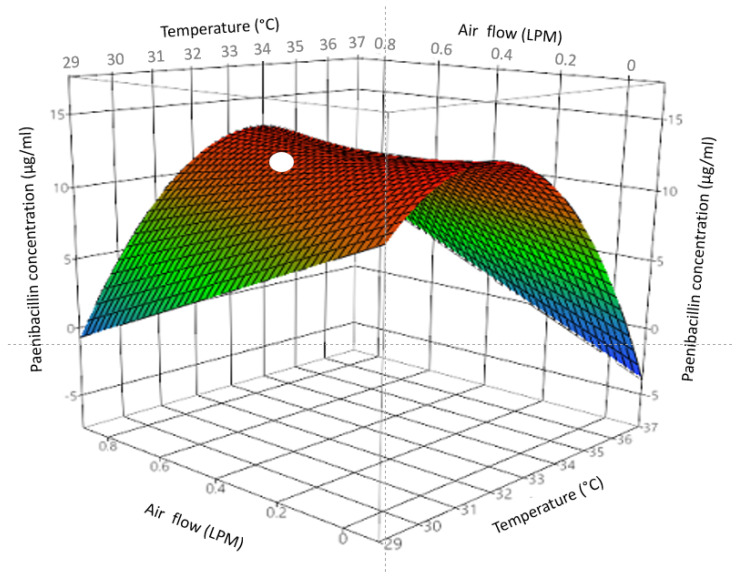
Response surface plot of paenibacillin titer (µg/mL) in the bulk media of bioreactors with no foam collection as a function of airflow (L/min; LPM) and temperature (°C). The symbol (
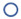
) indicates optimal conditions as determined by model at a temperature of 33 °C and airflow of 0.6 LPM.

**Figure 4 foods-11-02290-f004:**
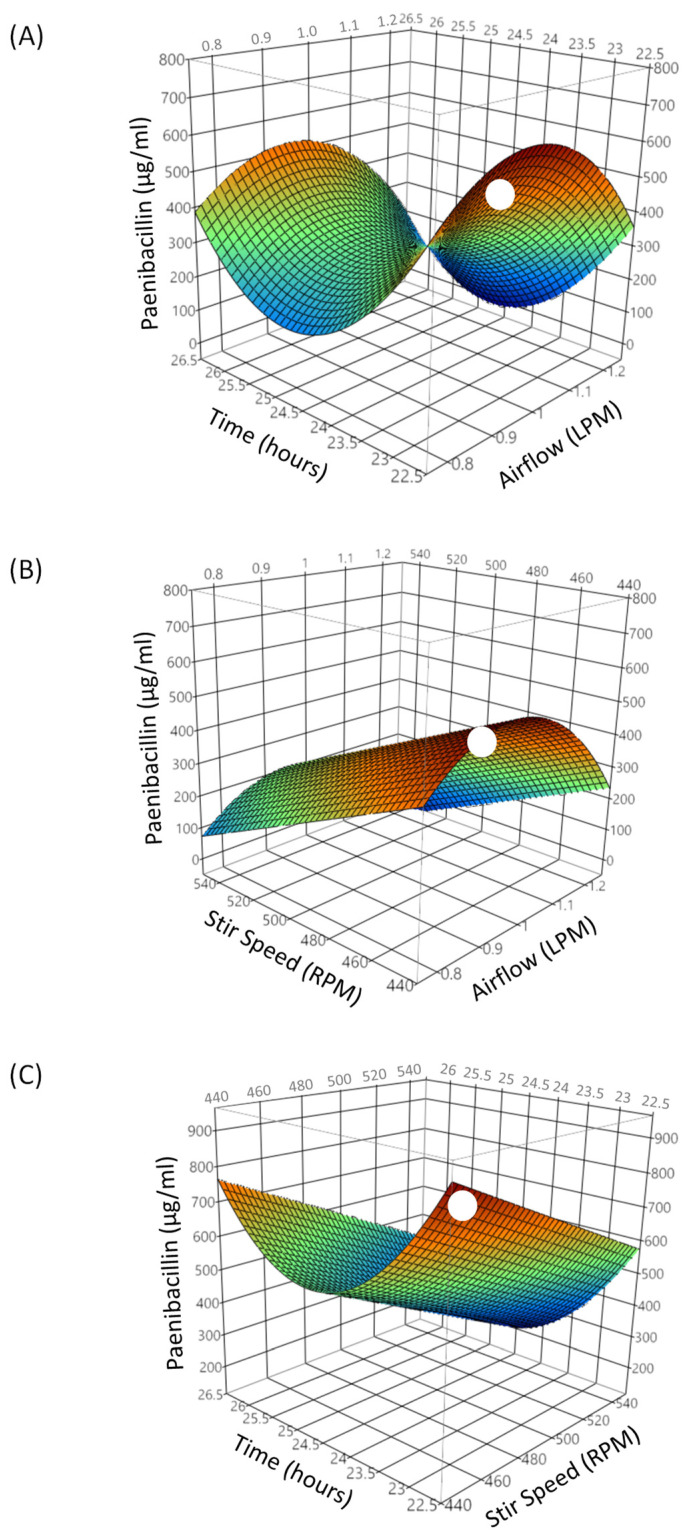
Response surface plots of paenibacillin titer (µg/mL) in foamate as a function of (**A**) time (h) and airflow (L/min; LPM), (**B**) airflow and stir speed (RPM), and (**C**) stir speed and time. The symbol (
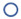
) indicates optimal conditions as determined by model at 0.95 LPM, incubation time of 23 h and stir speed of 450 RPM.

**Table 1 foods-11-02290-t001:** Initial partial factorial screening of parameters to test for paenibacillin concentration in bulk medium (in case bioreactions without foam collection) or foamate (in case bioreactions with foam collection) from a single bioreactor.

Temperature (°C)	Airflow (LPM ^a^)	Time (h)	Medium ^b^	Stir Speed (RPM ^c^)	Concentration of Paenibacillin (µg/mL) ^d^	
					Bulk medium	Foamate
30	0	16	0	500	16	<4.0
30	0	16	0	500	32	<4.0
30	0	16	0	300	16	<4.0
30	0	16	0	500	-	<4.0
30	0	24	1	300	-	<4.0
30	0.8	16	1	300	<4	<4.0
30	0.8	16	0	300	-	<4.0
30	0.8	16	0	500	-	<4.0
30	0.8	24	0	500	<4	470
30	0.8	24	1	300	16	310
30	0.8	24	0	300	<4	<4.0
30	0.8	24	0	500	-	200
36	0	16	1	500	<4	<4.0
36	0	16	1	300	4	<4.0
36	0	24	0	500	8	<4.0
36	0	24	0	300	-	<4.0
36	0.8	16	0	300	<4	<4.0
36	0.8	16	0	300	<4	<4.0
36	0.8	24	1	500	<4	250
36	0.8	24	1	300	4	<4.0

^a^ Liter per minute. ^b^ 0 = Tryptic soy broth, 1 = Tryptic soy broth + Yeast extract. ^c^ Revolutions per minute. ^d^ Detection limit of paenibacillin is 4 μg/mL foamate; any value < 4 μg/mL was entered in the model as zero.

**Table 2 foods-11-02290-t002:** Central composite design conditions tested to optimize concentration of paenibacillin in bulk media (bioreactions with no foam collection) for response surface methodology from a single bioreactor; other parameters were set to the lowest values.

Temperature (°C)	Airflow (L/min)	Paenibacillin Concentration (µg/mL) ^a^	Total Paenibacillin Yield (µg)
30	0	8	3200
30	0	8	3200
30	0	16	6400
30	0.4	8	3200
30	0.4	16	6400
30	0.8	4	1600
30	0.8	4	1600
30	0.8	4	1600
33	0	16	6400
33	0	16	6400
33	0.4	4	1600
33	0.4	4	1600
33	0.4	16	6400
33	0.8	16	6400
33	0.8	16	6400
36	0	<4	<1600
36	0	4	1600
36	0	4	1600
36	0	<4	<1600
36	0	4	1600
36	0	4	1600
36	0.4	8	3200
36	0.8	4	1600
36	0.8	8	3200
36	0.8	8	3200
36	0.8	16	6400

^a^ Detection limit of paenibacillin is 4 μg/mL; any value < 4 μg/mL was entered in the model as zero.

**Table 3 foods-11-02290-t003:** Central composite design conditions tested to optimize concentration of paenibacillin in foamate for response surface methodology from a single bioreactor ^a^; other parameters were set to the lowest values.

Airflow (L/min)	Time (h)	Stir Speed (RPM) ^b^	Paenibacillin Concentrationin Foamate ^c^ (µg/mL)	Bioreactor Used	Volume of Foamate Collected (mL)	Total Paenibacillin Yield (µg)
0.8	23	450	768	1	16.5	12,700
0.8	24	500	32	2	9.0	290
0.8	24	540	107	1	11.5	1230
0.8	25	520	53	2	8.25	440
0.8	26	450	192	2	7.75	1490
0.8	26	500	341	2	7.25	2470
0.8	26	540	171	1	10	1700
1	24	520	43	2	14	600
1	25	500	341	1	12.5	4260
1	25	500	256	2	14.5	3710
1	25	520	43	2	11	470
1	25	520	192	2	28	5400
1	25	520	43	2	8.25	360
1	25	540	171	1	5.5	940
1	26	520	107	2	20	2100
1.2	23	450	96	2	10	960
1.2	23	540	384	1	22	8400
1.2	24	500	28	2	12	340
1.2	24	540	64	1	7.5	480
1.2	25	520	26	1	12	310
1.2	26	450	512	1	14.5	7420
1.2	26	500	<4	2	12.5	<50
1.2	26	540	128	1	7.25	928

^a^ In all runs, paenibacillin concentration in bulk media was < 4.0 µg/mL; ^b^ Revolutions per minute; ^c^ Concentration lower than paenibacillin detection limit of 4 μg/mL was entered in the model as zero.

**Table 4 foods-11-02290-t004:** Experimental design for determining optimum conditions for maximum paenibacillin titer in bioreactors with or without foam collection.

Bioreaction	Stage of Bioreaction	Parameters Tested	Values	Significance (*p* Value)	Optimum
No foam collection	Partial factorial screening	Medium ^a^	TSB	>0.05	-
			TSB-YE		
		Airflow (L/min)	0.0	**<0.05**	-
			0.8		
		Stir speed (RPM) ^b^	300	>0.05	-
			500		
		Incubation temperature (°C)	30	**<0.05**	-
			36		
		Incubation time (h)	16	>0.05	-
			24		
	Central Composite Design	Airflow (L/min)	0.0	**<0.05**	0.6
			0.4		
			0.8		
		Incubation temperature (°C)	30	**<0.05**	33
			33		
			36		
		Maximum titer observed (µg/mL)			16
Foam collection	Partial factorial screening	Medium ^a^	TSB	>0.05	-
			TSB-YE		
		Airflow (L/min)	0.0	**<0.05**	-
			0.8		
		Stir speed (RPM) ^b^	300	**<0.05**	-
			500		
		Incubation temperature (°C)	30	>0.05	-
			36		
		Incubation time (h)	16	**<0.05**	-
			24		
	Central Composite Design	Ai flow (L/min)	0.8	**<0.05**	0.95
			1.0		
			1.2		
		Stir speed (RPM) ^a^	450	**<0.05**	450
			500		
			520		
			540		
		Incubation time (°C)	23	**<0.05**	23
			24		
			25		
			26		
		Maximum titer observed (µg/mL)			768

^a^ TSB; Tryptic soy broth, TSB-YE; Tryptic soy broth + Yeast extract; ^b^ Revolutions per minute.

## Data Availability

Not applicable.

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
