# Peer review of "Maximizing Recovery of Paenibacillin, a Bacterially Produced Lantibiotic, Using Continuous Foam Separation from Bioreactors"

_foods, 2022, doi:10.3390/foods11152290_

Round 1

Reviewer 1 Report

Information in this paper is of great interest considering the higher need for new antimicrobial compounds. Nature has a great capacity of producing biomolecules, but is our duty to find ways to produce them at a larger scale. In this regard, the manuscript describ a very interesting methodology of obtaining higher paenibacillin concentrations using   Paenibacillus polymyxa, by a reconfiguration of the lab bioreactor in order to collect the foam during the fermentation process. For optimal cultivation parameters, authors used response surface methodology in order to establish the amplitude of the paenibacillin accumulation at specific temperature, RPM and aeration variations

The authors did a great job in the lab and also during manuscript writing but I would like to make very few suggestions that could improve the overall presentation of the work.

L11 – specie name in italic

 L16-18 “Experiments proved that the significant parameters and their optimal condi-16 tions for antimicrobial concentration in the foam were: incubation at 30°C for 23h with airflow of 17 0.95 LPM, and agitation speed of 450 RPM.” Could be stated in a more comprehensive way. Rephrase.

 L21 in my opinion will be much easier to follow the paper if the authors will stay on ways of estimate the panibacillin: one way with µg/ml which is the concentration and the second way of quantitative estimation as yield of paenibacillin in the supernatant or in the entire cultivation batch (foam + supernatant)

The yield of paenibacillin in the bioreaction supernatant was estimated at X µg, like you did in the paper body.

L82 Maybe will be better if you will use the full name of the bacterial specie, since here is actualy the first time when you use it in the description of you experiments and in order not to be discriminatory for the second strain that you use, for wich you used full name. On the other hand in order not to have such high resemblence between present manuscript and the previous one of Campbell (reference nr.6), my suggestion is to blend the strains and cultivation media, somehow in order to encreas the diversity of the pharagraph, and name it Microbial strains cultivation, or something similar.

L110 I love the figure, but could you specify if there is a particular level difference between the bioreactor lead and the level of the foam collection flask?

Regarding preparation steps of bioreactor, you didn’t metioned if the equipment was autoclaved prior inoculation. Generaly, yes, but there are some experiments without sterilization step, but I think that your case must be in a steril environment.

L200-201 Remove the bioreactor equipment provider, you already sateted in the 2.1 Bioreacter setup. In the Table 3. There is “bioreactor used” followed by 1 or 2. I have to admit that I can’t understand the relevance.

Comments: I know that in biotechnology, researchers use seeding but in microbiology the most frecvent term is inoculation, so you can use

I read the previous paper were the authors presented some data regarding OD600, wich in my oppinien is quite an important parameter, especialy in biotehnology or up-scale experimets. Is obvious that if the bactrial cell density is low, the production of any peptide or compound will be on the small level, but if the cell density is at the highest level that  bacterial strain is capable of, the productivity will be acording.

So, my sugestion is that the reader of this paper needs few information regarding cell density and how the optimazed parameters influenced the culture.

When I read the manuscript, I was continously bothered by the abundance of the word “bioreaction” wich appers more then 50 times. If we are thinking at the meaning of the word, it has been used with a lot of meaning. From my point of view you can replace it with several others synonimes, that could be suggested by the team microbiologist.

In the manuscript authors used “antimicrobial” but in most, not in all cases is used incorectly since the term is regarding a property, is actualy an adjective, so it might be followed by a noun. in this case antimicrobial compound, antimicrobial peptide, substance…. I am awear that I am not a native english speaker, is understandable for you to used shorten version of some expresions...

Reviewer 2 Report

The research works focusing on optimising production and recovery of paenibacillin using continuous foam separation from bioreactors. This manuscript is nicely written. No spelling error was detected. The author manage to present their research finding clearly.

Materials & method

-          Suggest to add separate section for statistical analysis. Include type of software use to run RSM

Results and discussions

-          Subtopic 3.1 : Please include the conditions of constant parameter for foam and non-foam collection use

-          Line 269: What was ‘antimicrobial’ term in subtopic title was referring to? If it’s for penicillin, I suggest replacing it with penicillin or add penicillin next to antimicrobial.

Overall

The manuscript is nicely written, the objective and text construction were good.
